# Analysis of the Mitochondrial *COI* Gene and Genetic Diversity of Endangered Goose Breeds

**DOI:** 10.3390/genes15081037

**Published:** 2024-08-06

**Authors:** Hao Wu, Shangzong Qi, Suyu Fan, Haoyu Li, Yu Zhang, Yang Zhang, Qi Xu, Guohong Chen

**Affiliations:** 1College of Animal Science and Technology, Yangzhou University, Yangzhou 225009, China; wh18755049672@163.com (H.W.); mz120221524@stu.yzu.edu.cn (S.Q.); fansuyu20030729@163.com (S.F.); m19962648891@163.com (H.L.); yuzhang@yzu.edu.cn (Y.Z.); xuqi@yzu.edu.cn (Q.X.); ghchen2019@yzu.edu.cn (G.C.); 2Key Laboratory for Evaluation and Utilization of Livestock and Poultry Resources (Poultry), Ministry of Agriculture and Rural Affairs, Yangzhou 225009, China

**Keywords:** endangered goose, *COI* gene, genetic diversity, evolutionary analysis

## Abstract

The mitochondrial cytochrome c oxidase subunit I (*COI*) genes of six endangered goose breeds (Xupu, Yangjiang, Yan, Wuzong, Baizi, and Lingxian) were sequenced and compared to assess the genetic diversity of endangered goose breeds. By constructing phylogenetic trees and evolutionary maps of genetic relationships, the affinities and degrees of genetic variations among the six different breeds were revealed. A total of 92 polymorphic sites were detected in the 741 bp sequence of the mtDNA *COI* gene after shear correction, and the GC content of the processed sequence (51.11%) was higher than that of the AT content (48.89%). The polymorphic loci within the populations of five of the six breeds (Xupu, Yangjiang, Yan, Baizi, and Lingxian) were more than 10, the haplotype diversity > 0.5, and the nucleotide diversity (Pi) > 0.005, with the Baizi geese being the exception. A total of 35 haplotypes were detected based on nucleotide variation among sequences, and the goose breed haplotypes showed a central star-shaped dispersion; the FST values were −0.03781 to 0.02645, The greatest genetic differentiation (FST = 0.02645) was observed in Yan and Wuzong breeds. The most frequent genetic exchange (Nm > 15.00) was between the Wuzong and Yangjiang geese. An analysis of molecular variance showed that the population genetic variation mainly came from within the population; the base mismatch differential distribution analysis of the goose breeds and the Tajima’s D and Fu’s Fs neutral detection of the historical occurrence dynamics of their populations were negative (*p* > 0.10). The distribution curve of the base mismatches showed a multimodal peak, which indicated that the population tended to be stabilised. These results provide important genetic information for the conservation and management of endangered goose breeds and a scientific basis for the development of effective conservation strategies.

## 1. Introduction

Goose breeding in China has a long history, dating back 3000 years, and deep cultural significance [1,2,3]. To protect the diversity of rare and endemic endangered goose breeds, a national resource census showed that some local goose breeds, such as the Lingxian White (LX), Yangjiang (YJ), Yan (YE), Wuzong (WZ), Bai Zi (BZ), and the Xupu Goose (*Anser cygnoides*), have fewer than 1000 hens. At the same time, the efficiency of genetic resources and economic output are low in some places, often relying on the introduction of high-yielding breeds for crossbreeding. Chaotic crossbreeding is very serious, resulting in a decreasing number of purebred geese, and some existing goose breeds are at risk of extinction [4]. However, with industrialisation, urbanisation, and the impact of human activities, the population size of many domestic goose species has declined dramatically, posing a serious threat to their genetic diversity [5,6,7,8] and, ultimately, the survival of these traditional breeds. Protecting endangered goose species has become an urgent task; therefore, an analysis of the genetic diversity and phylogenetic relationships among goose breeds is essential for guiding conservation measures as well as crossbreeding selection [9,10,11]. The genetic diversity analysis of endangered goose breeds is crucial for the conservation and recovery of these breeds [12]. Genetic diversity, in terms of both gene frequency and the diversity of genotypes, plays crucial roles in the adaptability, evolution, and viability of a species. Mitochondrial DNA (mtDNA) is widely used in genetic diversity studies because it is highly variable between lineages, allowing researchers to establish different inheritance patterns [13,14,15]. The mitochondrial cytochrome c oxidase subunit I (*COI*) gene is a commonly used marker of mtDNA and plays an important role in species identification as well as population genetic structure analyses. By analysing the mitochondrial *COI* gene sequence, genetic differences and affinities between different populations can be revealed [16], providing important data for studying genetic diversity in endangered goose species.

The mitochondrial *COI* gene encodes a protein in mitochondrial DNA that varies between species and can be used to distinguish genetic differences between species [17]. Comparing and analysing the sequences of the mitochondrial *COI* gene can be helpful in investigating the genetic relationship among closely related species and infering evolutionary history and the kinship relationship between species. mtDNA is widely used in avian genetics studies because of its high conservation and rapid rate of evolution, making it an ideal marker for studying genetic diversity between species and within populations. The *COI* gene is commonly used as a standard DNA barcode sequence for the identification of animal species and can be informative in studying the evolutionary history of a species and detecting any phylogeographic structuring in a population.

The aim of this study was to analyse the genetic diversity of endangered goose breeds using mitochondrial *COI* gene sequences. Previous studies have mostly focused on the mitochondrial *D-loop* and *ND6* regions. However, the *COI* region has been little studied in geese. Deef et al. suggested that the mitochondrial *COI* region could be amplified to identify goose breeds [18]. There have been no systematic studies on the genetic diversity and evolutionary analysis of mitochondrial *COI* genes in the six locally endangered goose breeds (*Anser cygnoides*) included in our study. We collected samples from six different endangered breeds with different geographical distributions and sequenced the *COI* gene using sequencing technology (Sanger sequencing method). We then carried out analyses to infer the phylogeny, estimate the genetic diversity indices, and reveal the population genetic structure [19]. Through these analyses, we detected the level of genetic diversity among the different endangered goose breeds [20] and provide a scientific basis for their conservation and management. The results of this study will contribute to a better understanding of the genetic diversity of endangered goose breeds and provide important genetic information for their conservation. By protecting and increasing the genetic diversity of endangered breeds, their adaptability and survivability can be improved, thus effectively conserving these valuable biological resources. This study will also provide a reference and information for genetic diversity research on other endangered populations, which has theoretical and practical significance. Therefore, the analysis of the genetic diversity of endangered geese has important scientific significance and conservation value.

## 2. Materials and Methods

### 2.1. Test Animals

The following 300-day-old endangered goose breeds were used in this study. We collected 120 specimens from the conserved population (Figure 1), which were divided into six groups of 20 geese each according to species (*n* = 20/group, 10 ♂ + 10 ♀). They were Xupu (XP), Yangjiang (YJ), Baizi (BZ), Wuzong (WZ), Yan (YE) and Lingxian White Goose (LX) (Table 1 and Appendix A). All experimental geese were hatched and reared at the National Waterfowl Genetic Reserve (Taizhou, China).

### 2.2. DNA Extraction

A 1 to 2 mL sample of blood was collected from a subwing vein of each bird, placed in a blood collection tube containing EDTA anticoagulant, and stored at 4 °C. A genomic DNA extraction kit (Tiangen Biochemical Technology Co., Ltd., Beijing, China, DP304) was used to extract the DNA from the blood; the DNA was then diluted to 50 ng·µL^−1^ using a Biodropsis Ultra-micro Nucleic Acid and Protein Analyser and agarose gel electrophoresis. The extracted DNA was stored in a refrigerator at −20 °C until it was used in the analysis. 

### 2.3. Primer Design, Gene Amplification, and Sequence Determination

Based on the published full mtDNA sequence of goose (NCBI GenBank: MN122908.1), specific primers for the amplification of goose *COI* gene were designed using Primer 5.0 software (Figure 2). The primer sequences were as follows F: 5′ CCGCTCACGCCTTTGT 3′, R: 5′ CGGTAGGGATGGCAATG 3′. The primers were synthesised by Kengke Bioengineering Co. (Tokyo, Japan). The PCR mix amplification consisted of 384.5 ng of DNA template, 1.5 µL of each of the upstream and downstream primers (10 pmol·µL^−1^), 12.5 µL of 2×ApexHF FS PCR Mastermix, and 25 µL of ultra-pure water. The PCR amplification parameters were set as 94 °C pre-denaturation for 30 s, followed by 35 cycles of 98 °C denaturation for 10 s, 55 °C annealing for 5–15 s, and 72 °C extension for 5 s. The PCR products were subjected to 1.0% agarose gel electrophoresis; through the utilisation of Gel Imagers, the purity, size, and brightness were measured (Gel Doc^TM^ EZ), with strict control of sample quality and concentration to ensure sample specificity. It was ensured that the sequencing depth and coverage meet the expected requirements, while paying attention to the quality value and mismatch rate. Multidimensional analyses of data were performed to ensure the comprehensiveness and accuracy of results, and the products were then sent to Jinke Bioengineering (Nanjing) Co to perform sequencing. 

### 2.4. Statistical Analysis

The *COI* gene sequences were clipped and corrected using DNAStar 5.0 [21] (maintaining sequence consistency). The six goose breeds were grouped and analysed, and *COI* gene sequences were compared for homology (with Tajima’s D and Fu’s Fs neutrality tests using DNASP 6.10 software) [22]. In the two neutrality tests, Fu’s Fs value and Tajima’s D value were used to evaluate the historical demographic expansion using 1000 simulated samples. Negative values for both tests indicate that the population has experienced expansion, whereas positive values indicate that the population has experienced a bottleneck [23]; the output parameters were as follows: polymorphic sites, parsimony information sites, haplotype number, haplotype diversity, nucleotide diversity, and base mismatch difference analysis. Based on the output base mismatch difference analysis values, the base mismatch distribution curves were plotted using GraphPad Prism 8 [24]. Genetic distances among the breeds and the maximum-likelihood tree were calculated through 1000 resamplings of bootstrap using MEGA 11 software under the Kimura 2- parameter evolutionary model [25]. Haplotype network relational maps of the six goose species were constructed using “Popart 1.7” software [26].

## 3. Results

### 3.1. DNA Extraction, COI Gene Amplification, and Sequencing Results

Genomic DNA was extracted from 120 blood samples from the 6 breeds (6 breeds × 20 individuals), which were analysed as positive when amplifying the *COI* fragment of mtDNA as a template. The PCR amplification products were loaded and visualised on a 1% agarose gel electrophoresis (Figure 3), as inferred by the comparison with the DNA ladder, which was consistent with the expected target fragment size, and the shape of the bands was clear and complete, indicating that they were not degraded and had good specificity. Furthermore, no additional bright bands were found in the sample lanes, indicating that the samples were not contaminated.

### 3.2. Population Genetic Diversity

#### 3.2.1. Locus Information and Nucleic Acid Diversity Analysis

Goose *COI* characterisation and nucleotide diversity are shown in Table 2. A homology comparison of *COI* gene sequences of the six breeds was carried out using DNA SP 6.0. The final length of the analysed sequences was 740 bp after the nucleotides were aligned, trimmed, and manually corrected, revealing the existence of 92 polymorphic sites, which accounted for approximately 12.43% of the total number of sequence sites (92/740), and the GC content of the sequences (51.11%) was found to be higher than the AT content (48.89%). The number of polymorphic loci in five of the six breeds—XP, YJ, YE, BZ, and LX—was greater than 10 (NPIS + PIS), while the haplotype diversity (Hd) was always greater than 0.95. Nucleotide diversity (Pi) was greater than 0.005 in all cases, except the Baizi group. The mean number of average nucleotide differences was much higher in XP goose than in the other five goose species, lowest in BZ goose, and similar in YJ and YE geese.

#### 3.2.2. Haplotype Distribution and Frequency Analysis

The results of the study show that the haplotypes of goose breeds had a centred star-like dispersion (Figure 4); in total, 35 haplotypes were detected according to nucleotide variation among the sequences. The distribution of the 35 haplotypes in the six populations is shown in Table 3. The distribution of various haplotype variable loci among the populations is shown in Figure 5, with haplotypes 4, 7, 12 and 15 being dominant. The distribution and frequency of the detected 35 haplotypes are shown in Table 4. The highest number of haplotypes was found in the Yan breed, followed by Xupu, with 19 and 18 haplotypes, respectively. The number of haplotypes was 17 in Yangjiang goose and Lingxian goose, and 15 in Wuzong goose and Baizi goose. Yangjiang and Baizi geese had the highest number of endemic haplotypes, accounting for 16.66% and 40.00% of their total haplotypes, respectively. In addition, haplotype 4 was common to all the analysed populations.

### 3.3. Genetic Variation among Populations

Gene flow, Nm, and genetic differentiation index, Fst, were calculated among the six populations using Arlequin 3.5 software (Table 5). In general, genetic differentiation among the populations was low. The results showed that the Fst values among the populations ranged from −0.03781 to 0.02645, with the smallest genetic differentiation (−0.03781) between the Baizi and Yangjiang goose populations and the largest (0.02645) between the Yan and Wuzong populations. The Nm values among the six populations ranged from −37.34 to 67.06, with the most frequent genetic exchanges occurring between the Wuzong and Yangjiang breeds. The analysis of molecular variance (AMOVA) showed that genetic variation was mainly from within populations, with a sum of squares of 3.916 between populations and 100.455 within populations (Table 6). The variance component was −0.00684 between populations and 0.91075 within populations. 

### 3.4. Clustering Relationship Analysis

To allow a closer examination of the genetic distances between the six goose breeds (Table 7), a phylogenetic map of the six breeds was constructed by using MEGA 11 software. The results of the model show that the six goose breeds in this study can be grouped into two clusters based on their respective mitochondrial DNA *COI* gene sequence (Figure 6). The LX, WZ, and XP breeds are aggregated into one group, and the YE, YJ, and BZ breeds are aggregated into another group. Lastly, these two main groups are then aggregated into a single large taxon.

### 3.5. Analysis of Historical Population Dynamics

Based on the mitochondrial *COI* gene sequences, DNASP software was used to analyse the neutrality test of the six breeds of geese. The results showed that the neutral tests of Tajima’s D value and Fu’s Fs value were negative and the difference was not significant *p* > 0.10 (Table 8), indicating that the six populations conformed to the result of neutral mutation in a stable state. The base differential mismatch analyses of the *COI* gene sequences of six populations (breeds) showed that the base differential mismatches were distributed in a multi-peak pattern (Figure 7); in other words, the distribution did not conform to the pattern of the single-peak curve of population expansion, which is consistent with the results of the neutrality test.

## 4. Discussion

The analysis of the genetic diversity of endangered goose breeds is an important step in the conservation and management of this biological resource [27,28,29,30]. In the present study, we analysed the genetic diversity of six endangered breeds of Chinese goose with different geographical distributions, using mitochondrial *COI* gene sequences. By sequencing and analysing the mitochondrial *COI* gene sequence, it was found that the GC (51.11%) content was higher than the AT content (48.89%), indicating a more pronounced base bias. In comparison, Zhuo et al. [31], also utilising the mitochondrial *COI* gene, showed that the GC content was lower than the AT content in the black-boned grouse. It has been speculated that in the course of the evolutionary process, the GC content of the genome may be affected by a variety of factors, including mutation rate, selection pressure, and genome size [32]; if a gene has been subjected to selective pressure for GC preference during evolution, then the GC content of that gene may be relatively high. Nucleotide diversity is commonly used to indicate the type and number of nucleotides in a segment of a DNA or RNA sequence [33] and can be used to measure the degree of variation or genetic diversity in a genome or a specific region of the genome. Higher nucleotide diversity indicates that more variation exists within that region, and the reverse indicates less variation. Haplotype diversity is often used to indicate the variety and number of haplotypes within groups of individuals. Haplotype diversity can provide information on population history, evolutionary processes, relatedness, and genetic health [34]. By analysing haplotype diversity, important information, such as genetic structure, migration history, population size, and genetic fitness within a population, can be revealed. The nucleotide difference number indicates the number of nucleotides that differ at the same position in two sequences. When comparing two DNA or RNA sequences, the nucleotide difference number can help determine the similarities and differences between them. A higher nucleotide difference number indicates a greater difference between the two sequences, whereas a lower number indicates a greater similarity between them. The mean haplotype diversity (Hd), nucleotide difference (K), and nucleotide diversity (Pi) of the six goose populations in the present study were 0.98, 5.637, and 0.00771, respectively. In contrast, the comparable values obtained in a study by Zhang et al. [35] involving Chinese domestic ducks originating from mallard (*Anas platyrhynchos*) and mottled duck (*A. zonorhyncha*) were as follows: Hd, 0.653; K, 3.104; and Pi, 0.005. These values are considerably lower than the comparable values found here with the six endangered goose breeds. This indicates that the six goose breeds we examined show high genetic diversity, which helps them adapt to different environmental conditions and resist external pressures, thus increasing the likelihood of long-term survival of these populations.

Our results showed a certain degree of genetic variation among different endangered goose breeds. A total of 35 haplotypes were detected based on the nucleotide variation between sequences; the greatest number of haplotypes was found in Yan goose. Yangjiang and Baizi geese had the highest number of endemic haplotypes. The endemic haplotypes in these breeds accounted for 35.29% and 40.00% of the total haplotypes, excluding haplotype 4, which was a population-shared haplotype among the six breeds. The construction of a phylogenetic tree showed that the study population as a whole was divided into six haplogroups, and the genetic relationships between different breeds were clearly evident [36]. This information is important to understand the kinship, population differentiation, and historical evolution of endangered goose breeds [37]. The results of this study also reveal the population genetic structures of the different breeds. By analysing the population genetic structure, we can understand the degree of gene flow and genetic differentiation among different populations [38]. The coefficient of genetic differentiation FST of a population reflects the proportion of genetic variation among populations. The results showed that the Fst values between populations were in the range of −0.03781 to 0.02645, indicating that there was little difference in gene frequency and a low degree of genetic differentiation within the population. Warzecha et al. conducted paired analysis of FST in geese (*Anser anser*) from abroad [39]. The FST values of Landes and Romanska populations were very high (0.261), significantly higher than those of Chinese goose breeds (*Anser cygnoides*). This indicates that endangered goose breeds do not have genetic differentiation, and the genetic structure between populations is completely identical. This may be due to random drift or low levels of selection pressure. The Nm values of the six populations ranged from −37.34 to 67.06, with the genetic variation between Wuzong goose and Yangjiang goose being the most frequent. This phenomenon may be due to the geographical proximity of the two varieties, coupled with artificial selective mating, resulting in gene flow. Gene flow between different subgroups or populations is more frequent. A multi-peaked mismatch distribution indicates that the shape of the gene tree is highly stochastic, indicating that the population dynamics are stable and balanced, whereas a single-peaked mismatch distribution indicates that the population has recently expanded or that the neighbouring populations have expanded their distributional areas with a high level of migration. In the present study, the distribution of nucleotide mismatches showed an irregular curve, which did not conform to the single-peak curve pattern of population expansion, and the values of Tajima’s D and Fu’s Fs did not reach the level of significance, indicating that the six goose breeds did not deviate from neutral selection and did not undergo a large-scale population expansion in the past. 

When conserving endangered goose breeds, the degree of genetic linkage and differentiation between populations must be considered to develop targeted conservation measures. Protecting the genetic diversity of endangered species is an important aspect of conservation [40]. Genetic diversity increases the adaptability and survivability of species and is crucial for their long-term survival and reproduction [41,42]. By understanding the genetic diversity of different endangered goose breeds, appropriate measures can be taken to protect and increase their genetic diversity [43]. For example, the population size and level of genetic diversity of endangered goose species can be increased through artificial propagation [44], population reconstruction [45], and habitat protection [46], thus improving their survival. By analysing the genetic diversity of endangered goose species using the mitochondrial *COI* gene, we can better understand their genetic status and provide a scientific basis for their conservation and management [47]. Protecting the genetic diversity of endangered goose species is of great significance for maintaining biodiversity and protecting the ecological balance, and the results of this study will provide important references and guidance for related conservation work. This study also provides a reference for studying the genetic diversity of other endangered species, which has both theoretical and practical significance. We hope that this study provides useful support and assistance for the conservation and management of endangered goose species.

## 5. Conclusions

Analyses of the genetic diversity of endangered goose species are important for their conservation and management. The sequence analysis of the mitochondrial *COI* gene in endangered goose species can reveal genetic differences and relationships between them. This can help determine genetic mobility between different populations, assess the genetic health of populations, and provide a scientific basis for the development of conservation measures. In addition, the genetic diversity analysis of the mitochondrial *COI* gene can help to identify the genetic characteristics of populations and provide important information for species identification and population genetic structure studies. This can help better understand the population dynamics and evolutionary history of endangered goose breeds and provide scientific guidance for their conservation and management. Therefore, by analysing the genetic diversity of the mitochondrial *COI* gene in endangered breeds, we can gain insights into the genetic characteristics and genetic structure of these breeds and provide important references for their conservation and management.

## Figures and Tables

**Figure 1 genes-15-01037-f001:**
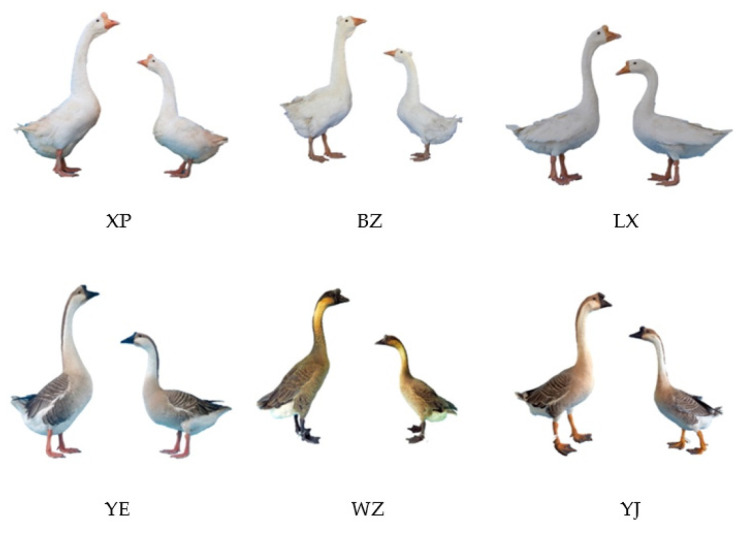
Goose breeds used in the study.

**Figure 2 genes-15-01037-f002:**
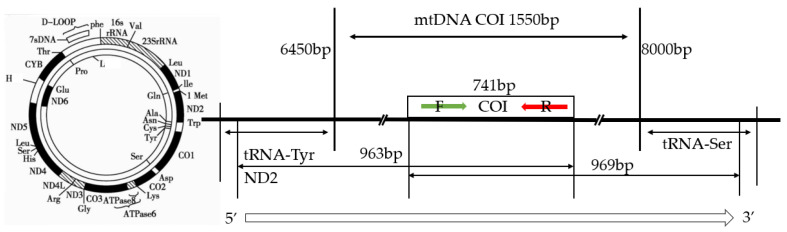
Design of the primers.

**Figure 3 genes-15-01037-f003:**
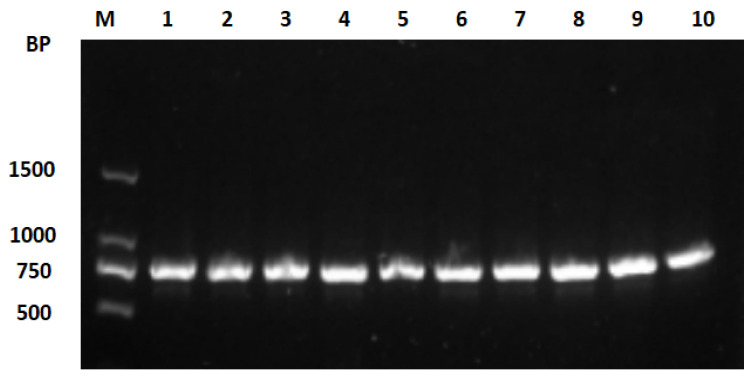
Example of standard pattern obtained for the analysis of the PCR products on 1% agarose gel electrophoresis.

**Figure 4 genes-15-01037-f004:**
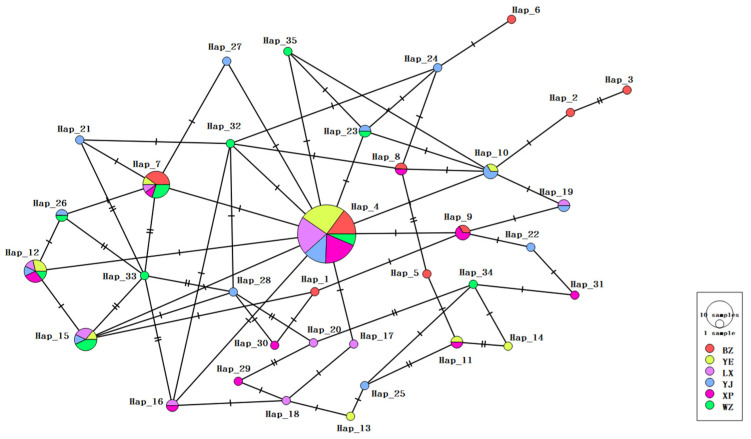
Median-joining network diagram constructed based on *COI* gene haplotypes. Each circle represents a unique haplotype, the colour represents endangered geese of different breeds, and the size of the circle is proportional to the number of isolates contained. The lines (shaded markers) on the branches indicate the location of the mutation, with one line for each mutation.

**Figure 5 genes-15-01037-f005:**
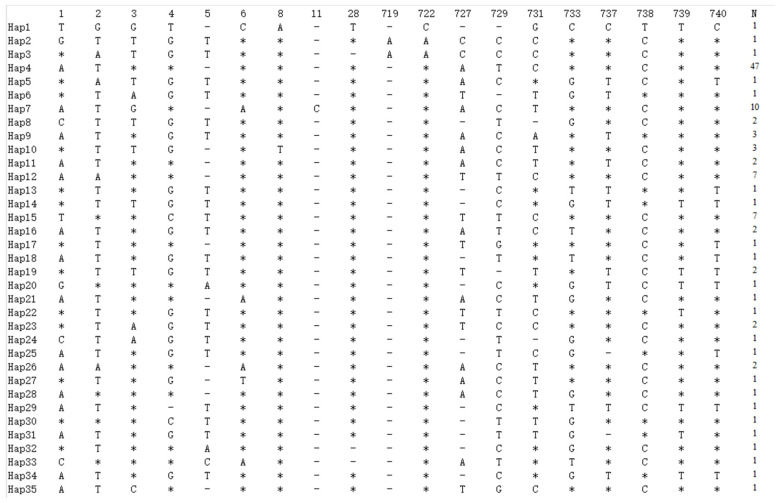
Distribution of variable loci based on haplotype control of the mtDNA *COI* gene. Note: “*” indicates base-identical sequences; “-” indicates a base deletion.

**Figure 6 genes-15-01037-f006:**
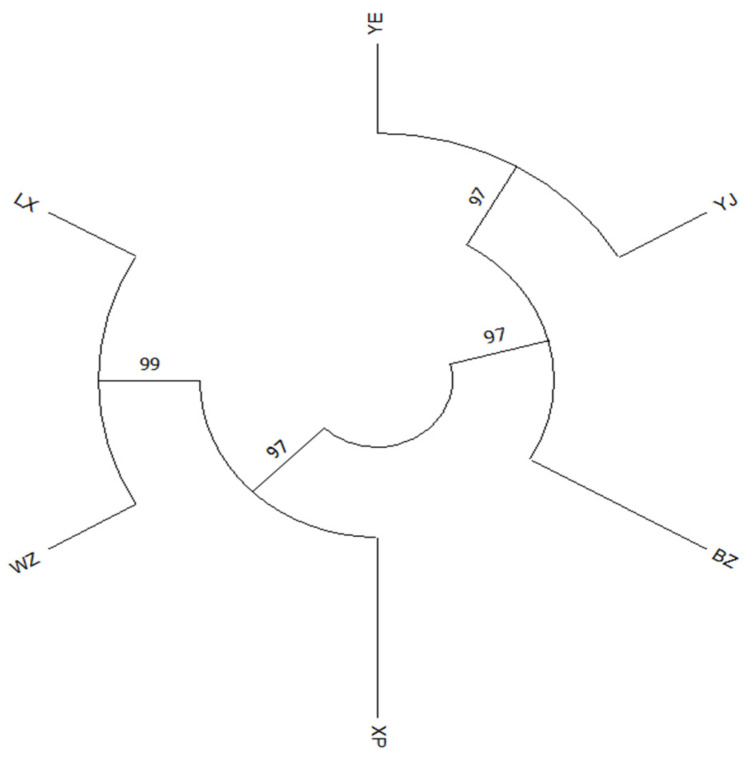
MEGA11 phylogeny (maximum-likelihood) map, constructed on the basis of mtDNA *COI* sequences from the six breeds of geese.

**Figure 7 genes-15-01037-f007:**
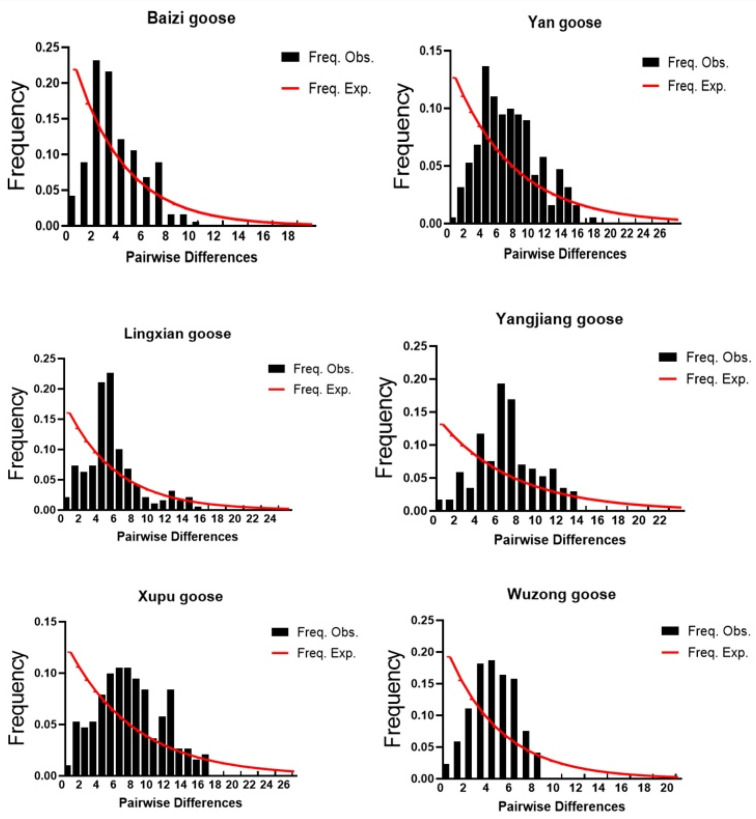
Distribution of base difference mismatches in the mt *COI* gene sequence of endangered goose breeds.

**Table 1 genes-15-01037-t001:** Details of the six breeds of geese.

Breed	Origin	Livestock Purpose	Physical Characteristics
XP	Xupu, Hunan	Meat and liver	Large body; oblong-cylindrical in shape
YJ	Yangjiang, Guangdong	Meat (small breed)	Compact body
BZ	Jinxiang, Shandong	Eggs	Medium-sized head
WZ	Qingyuan, Guangdong	Meat (small breed)	Compact body, ebony feathers
YE	Liuan, Anhui	Meat (medium-sized breed)	Large body; rectangular in shape
LX	Yanling, Hunan	Meat (small breed)	Small and compact body; cylindrical in shape

**Table 2 genes-15-01037-t002:** Endangered goose mtDNA *COI* locus information, haplotypes, and nucleotide diversity.

Breed	MS	NPIS	PIS	GC	K	Pi	Hd
XP	715	4	16	51.01%	7.305	0.0099 ± 0.00130	0.989 ± 0.019
YJ	716	4	12	50.62%	6.608	0.0090 ± 0.00088	0.982 ± 0.026
BZ	709	3	8	50.33%	3.568	0.0049 ± 0.00079	0.958 ± 0.033
WZ	711	2	7	50.54%	4.199	0.0058 ± 0.00041	0.977 ± 0.023
YE	718	4	15	50.71%	6.905	0.0093 ± 0.00110	0.995 ± 0.018
LX	717	3	14	50.93%	5.237	0.0071 ± 0.00110	0.979 ± 0.024

Note: NPIS: non-parsimony informative loci; PIS: parsimony informative sites; MS: monomorphic sites; GC: GC content; Hd: haplotype diversity; Pi: nucleotide diversity; K: average number of nucleotide differences.

**Table 3 genes-15-01037-t003:** Distribution of 35 haplotypes in the six breeds.

Haplotype	Breed and Number of Individuals	Total Number of Individuals with Haplotype
H1	[BZ1]	1
H2	[BZ1]	1
H3	[BZ1]	1
H4	[BZ7 YE12 LX10 YJ6 XP9 WZ3]	47
H5	[BZ1]	1
H6	[BZ1]	1
H7	[BZ4 YE1 LX1 XP1 WZ3]	10
H8	[BZ1 XP1]	2
H9	[BZ1 XP2]	3
H10	[YE1 YJ2]	3
H11	[YE1 XP1]	2
H12	[YE2 LX1 YJ1 XP2 WZ1]	7
H13	[YE1]	1
H14	[YE1]	1
H15	[YE1 LX2 YJ1 WZ3]	7
H16	[LX1 XP1]	2
H17	[LX1]	1
H18	[LX1]	1
H19	[LX1 YJ1]	2
H20	[LX1]	1
H21	[YJ1]	1
H22	[YJ1]	1
H23	[YJ1 WZ1]	2
H24	[YJ1]	1
H25	[YJ1]	1
H26	[YJ1 WZ1]	2
H27	[YJ1]	1
H28	[YJ1]	1
H29	[XP1]	1
H30	[XP1]	1
H31	[XP1]	1
H32	[WZ1]	1
H33	[WZ1]	1
H34	[WZ1]	1
H35	[WZ1]	1

**Table 4 genes-15-01037-t004:** Number of haplotypes and frequency of distribution in endangered goose breeds.

Breed	Haplotype Number	Number of Unique Haplotypes	Unique Single/Double Type Frequency
WZ	15	4 (Hap32, Hap33, Hap34, Hap35)	26.66%
XP	18	3 (Hap29, Hap30, Hap31)	16.66%
YJ	17	6 (Hap21, Hap22, Hap24, Hap25, Hap27, Hap28)	35.29%
LX	17	3 (Hap17, Hap18, Hap20)	17.64%
YE	19	2 (Hap13, Hap14)	10.52%
BZ	15	6 (Hap1, Hap2, Hap3, Hap4, Hap5, Hap6)	40.00%

**Table 5 genes-15-01037-t005:** Genetic differentiation index Fst (lower diagonal) and gene flow Nm (upper diagonal) among endangered goose breeds; “inf” is infinity.

Goose Breed	BZ	YE	LX	YJ	XP	WZ
BZ		−37.34	23.25	−29.21	50.14	Inf
YE	−0.01349		−14.18	−28.50	−15.46	24.93
LX	0.02105	−0.03072		4.77	Inf	Inf
YJ	0.03781	−0.02864	−0.00084		Inf	67.06
XP	0.00087	−0.03115	−0.00349	−0.02390		18.40
WZ	0.00079	0.01966	0.00869	−0.00726	0.02645	

**Table 6 genes-15-01037-t006:** AMOVA of the six breeds.

Source of Variation	Degrees of Freedom	Square Sum	Variance Component	Percentage of Variation
Interpopulation	5	3.916	−0.00684	−0.76
Intrapopulation	106	96.539	0.91075	100.76
Total	111	100.455	0.90391	100.00

**Table 7 genes-15-01037-t007:** Genetic distances between the six breeds.

Breed	BZ	YE	LX	YJ	XP	WZ
BZ		0.01002	0.00988	0.01021	0.01013	0.01110
YE	0.01002		0.00918	0.00963	0.00965	0.01054
LX	0.00988	0.00918		0.00920	0.00938	0.01020
YJ	0.01021	0.00963	0.00920		0.00989	0.01076
XP	0.01013	0.00965	0.00938	0.00989		0.01075
WZ	0.01110	0.01054	0.01020	0.01076	0.01075	

**Table 8 genes-15-01037-t008:** Analysis of historical population dynamics of endangered goose breeds.

Metric	BZ	YE	LX	YJ	XP	WZ
Fu’s Fs	−9.0600	−12.5530	−10.0230	−9.1800	−9.4990	−8.4730
*p* value	*p* > 0.10	*p* > 0.10	*p* > 0.10	*p* > 0.10	*p* > 0.10	*p* > 0.10
Tajima’s D	−2.5156	−0.0777	−0.5958	−0.2694	−0.4158	−0.1789
*p* value	*p* > 0.10	*p* > 0.10	*p* > 0.10	*p* > 0.10	*p* > 0.10	*p* > 0.10

## Data Availability

The original contributions presented in the study are included in the article/Appendix A, further inquiries can be directed to the corresponding author.

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
