# Peer review of "Analysis of the Mitochondrial COI Gene and Genetic Diversity of Endangered Goose Breeds"

_genes, 2024, doi:10.3390/genes15081037_

Round 1

Reviewer 1 Report

Comments and Suggestions for Authors

Comments

  1. Author should provide more background on the significance of the study and its relevance to conservation efforts in the introduction part. Also, explain why these specific goose breeds were chosen and their ecological or economic importance.
  2. Author should clarify the research gaps this study aims to fill.
  3. Mention the criteria for selecting the 20 individuals from each breed to ensure representativeness in Line 82.
  4. Give a brief explanation of each statistical test (e.g., Tajima's D, Fu's Fs) in line 115-123 and why they were used. This will help readers understand the relevance of these tests to the study's objectives.
  5. Include a brief description of the quality control measures taken during sequencing to ensure data accuracy in Line 128-134.
  6. Line 139-149, Author should clearly define the terms used (e.g., polymorphic sites, haplotype diversity) and their relevance to genetic diversity studies.
  7. Author should strengthen the study by comparing their results with those of similar studies. Also, highlight any novel findings and their implications for conservation.
  8. Cite the references for all the software used in the manuscript.
Comments on the Quality of English Language

English editing is required

Reviewer 2 Report

Comments and Suggestions for Authors

The manuscript is structured correctly and sequentially and is easy to read. The topic covered is quite specific and provides useful ideas for the protection of genetic variability in the geese populations analysed. I believe that the M&M section needs to be more streamlined and clear and that the results can be improved (see comments and suggestions in the attached file). Furthermore, in the discussion I highlighted some points that can be explored further. In general, I noticed that the same concept is often repeated and that in some cases the sentences should be moved from one section to a more relevant one.

Round 2

Reviewer 1 Report

Comments and Suggestions for Authors

The authors addressed all my concerns; therefore, I recommend it to accept for publication. 

Comments on the Quality of English Language

Minor English editing is required. 

Reviewer 2 Report

Comments and Suggestions for Authors

Dear Editor, Dear Authors,
It seems that my 96 comments included in my first revison have not been seen by the Authors. They didn’t check the comments section of the PDF file where each revised statement to be use in substitution of the previous one were reported. The Authors just removed the sentences without including the revision suggested. For this reason, some statements/paragraphs don’t make sense. I recommend the Authors to address this issue, they will find the revised statements/paragraphs and the suggestions to improve the main text in my first revision (genes-3114962-peer-review-v1-1_Rev.pdf)
Some of the statements/section that still need to be revised, according to the comment section are (line numbering referred to the first revision):
Line 46:  here I suggested to replace “COI gene” with “cytochrome c oxidase subunit I (COI) gene” while the Authors just removed “COI gene” without replacing it with nothing.
The other revision to be completed are:
Lines 53-55
Line 65
Lines 66-68
Line 69
Line 103 (here I asked to add the sequences of 2F and 1R primers and to explain why you needed to use an internal primer pair)
Lines 109-11
Line 115
Lines 128-135
and so on.
Please, keep going through the whole manuscript and complete all the revisions!

Author Response

Dear reviewer:   Thank you for your feedback and detailed explanation of my manuscript. I understand the issue you mentioned, and indeed the 96 comments included in the first revision were not fully read, resulting in some missing paragraphs and statements.   We have carefully reviewed the comments section in the PDF file and made revisions to effectively understand and implement each suggestion.   Thank you again for your advice and patient guidance.
